# Specific Upregulation of TRPC1 and TRPC5 Channels by Mineralocorticoid Pathway in Adult Rat Ventricular Cardiomyocytes

**DOI:** 10.3390/cells9010047

**Published:** 2019-12-23

**Authors:** Fiona Bartoli, Soraya Moradi Bachiller, Fabrice Antigny, Kaveen Bedouet, Pascale Gerbaud, Jessica Sabourin, Jean-Pierre Benitah

**Affiliations:** 1Inserm, UMR-S 1180, Signalisation et Physiopathologie Cardiovasculaire, Université Paris-Saclay, 92296 Châtenay-Malabry, France; F.Bartoli@leeds.ac.uk (F.B.); soraya_22mb@hotmail.com (S.M.B.); kaveen.bedouet@gmail.com (K.B.); pascale.gerbaud@u-psud.fr (P.G.); 2Inserm, UMR-S 999, Centre Chirurgical Marie Lannelongue, Université Paris-Saclay, 92350 Le Plessis-Robinson, France; fabrice.antigny@u-psud.fr

**Keywords:** aldosterone, TRPC channels, STIM1, adult rat ventricular cardiomyocytes, SOCE, *I*_SOC_

## Abstract

Whereas cardiac TRPC (transient receptor potential canonical) channels and the associated store-operated Ca^2+^ entry (SOCE) are abnormally elevated during cardiac hypertrophy and heart failure, the mechanism of this upregulation is not fully elucidated but might be related to the activation of the mineralocorticoid pathway. Using a combination of biochemical, Ca^2+^ imaging, and electrophysiological techniques, we determined the effect of 24-h aldosterone treatment on the TRPCs/Orai-dependent SOCE in adult rat ventricular cardiomyocytes (ARVMs). The 24-h aldosterone treatment (from 100 nM to 1 µM) enhanced depletion-induced Ca^2+^ entry in ARVMs, as assessed by a faster reduction of Fura-2 fluorescence decay upon the addition of Mn^2+^ and increased Fluo-4/AM fluorescence following Ca^2+^ store depletion. These effects were prevented by co-treatment with a specific mineralocorticoid receptor (MR) antagonist, RU-28318, and they are associated with the enhanced depletion-induced N-[4-[3,5-Bis(trifluoromethyl)-1H-pyrazol-1-yl]phenyl]-4-methyl-1,2,3-thiadiazole-5-carboxamide (BTP2)-sensitive macroscopic current recorded by patch-clamp experiments. Molecular screening by qRT-PCR and Western blot showed a specific upregulation of TRPC1, TRPC5, and STIM1 expression at the messenger RNA (mRNA) and protein levels upon 24-h aldosterone treatment of ARVMs, corroborated by immunostaining. Our study provides evidence that the mineralocorticoid pathway specifically promotes TRPC1/TRPC5-mediated SOCE in adult rat cardiomyocytes.

## 1. Introduction

So far, the physiological impact of store-operated Ca^2+^ entry (SOCE) in adult cardiomyocytes is still under debate, notably due to low-to-moderate expression of their molecular constituents including the molecular complex of STIM1 with Orai1 channels but also with different isoforms of transient receptor potential canonical channels (TRPCs; notably 1, 4, and 5) [1,2,3,4,5]. However, there is now strong evidence revealing a role for SOCE in developmental and pathologic cardiac growth, in which Ca^2+^ handling regulation through Ca^2+^-permeable channels is a cornerstone [6,7,8,9,10,11,12]. Indeed, hardly detectable in healthy adult cardiac myocytes [13,14], SOCE is present in embryonic and neonatal cardiac myocytes [15] and re-emerges during cardiac hypertrophy and heart failure [1,3,4,16]. While it was proposed that the upregulation of cardiac SOCE during pathological growth is part of the fetal gene program [4,17], neuroendocrine factors, which underpin the pathogenesis of heart failure and trigger Ca^2+^-dependent processes leading to cell growth and cardiac hypertrophy, might be involved. Notably, in addition to being an Na^+^-retaining and kalituretic hormone that acts on the kidney, the adrenal steroid aldosterone and its classical biological effector, the mineralocorticoid receptor (MR), exert deleterious effects on cardiac function, independently of their action on blood pressure [18]. Indeed, cardiomyocytes express the MR, a ligand-dependent transcription factor interacting with the glucocorticoid-responsive elements of target genes, whose activity might control Ca^2+^ homeostasis on the target cells [19,20,21]. Prior studies suggested that stimulation of neonatal rat ventricular myocytes (NRVMs) or human embryonic stem-cell-derived cardiomyocytes with angiotensin II, phenylephrine, endothelin-1, or aldosterone evoked an exacerbated SOCE, correlating with increased expression of Orai1, TRPC1, TRPC4, and/or TRPC5 [11,12,22,23,24,25,26]. Arguably, primary cultures of NRVMs constitute a reliable in vitro model to study time-related myocardial cell biology but may not accurately reflect the matured adult state. Here, we showed in adult rat ventricular cardiomyocytes that the aldosterone-enhanced SOCE is associated with specific upregulation of TRPC1/C5, pointing out the necessity to corroborate data obtained in NRVMs in a mature system.

## 2. Material and Methods

All data and materials supporting the findings of this study are available within the article or from the corresponding authors upon reasonable request.

All experiments were carried out according to the ethical principles laid down by the French Ministry of Agriculture (agreement D-92–019-01) and were performed in accordance with the guidelines from Directive 2010/63/EU of the European Parliament on the protection of animals. Animal studies were approved by the local Ethics Committee (CREEA Ile-de-France Sud).

### 2.1. Chemicals

Nifedipine (NIF) and caffeine (caf) were obtained from Sigma-Aldrich. The RU-28318, thapsigargin (Tg) and the KB-R7943 were obtained from TOCRIS Bioscience (Bristol, UK). Fluo-4/AM and Fura-2/AM were obtained from Thermo Fisher Scientific (Waltham, MA, USA).

### 2.2. Ventricular Myocyte Isolation

Adult rat ventricular cardiomyocytes (ARVMs) were isolated from rat hearts using a standard enzymatic perfusion method and maintained for 24 h at 37 °C in a serum-free Tyrode solution (in mM: 130 NaCl, 5.4 KCl, 0.4 NaH_2_PO_4_, 0.5 MgCl_2_, 25 HEPES, 22 d-glucose, 1 CaCl_2_; pH 7.4 with NaOH) with or without aldosterone and with or without RU28318, as previously described [19]. Only rod-shaped cells, quiescent when unstimulated and excitable, were used. In comparison to freshly isolated ARVMs, no major alterations in cellular excitability and size were noted [19,20,21,27].

### 2.3. Measurement of Cytosolic Ca^2+^ Changes

After washing, the ARVMs were incubated at room temperature in a solution containing (in mM) 135 NaCl, 5 KCl, 1.8 CaCl_2_, 1 MgCl_2_, 10 HEPES, 10 d-glucose, pH 7.4 with NaOH, supplemented either with 5 µM Fluo-4/AM (for 30 min) or with 1 μM Fura-2/AM (for 15 min). Both fluorescence probes were solved in DMSO plus 20% pluronic acid in stock solutions. The SOCE was measured as described previously [14]. The Fura-2 340/380 ratios were calibrated using the equation [Ca^2+^]_I_ = K_D_·β·(R − R_min_)/(R_max_ − R), where R is the fluorescence ratio recorded at the two excitation wavelengths (F_340_ and F_380_), K_D_ represents the dissociation constant, R_min_ and R_max_ are the fluorescence ratios under Ca^2+^-free and Ca^2+^-saturating conditions, and β = F_340_, zero Ca^2+^/F_380_, saturating Ca^2+^.

For in situ measurements of R_min_ and R_max_, Fura-2-loaded ARVMs were firstly exposed to Ca^2+^-free solution with 10 mM caf + 5 µM Tg for 5 min, to empty Sarcoplasmic Reticulum (SR) Ca^2+^ stores. The bath solution was then switched to K^+^ buffer (in mM: 10 NaCl, 130 KCl, 1 MgCl_2_, 10 HEPES, pH: 7.2, ionic strength: 0.142, 37 °C) containing 10 mM Ethylene glycol-bis(2-aminoethylether)-*N*,*N*,*N*′,*N*′-tetraacetic acid (EGTA). To obtain R_min_, the cells were incubated for 30 to 45 min with 20 µM non-fluorescent Ca^2+^ ionophore, ionomycin, and 10 µM carbonyl cyanide *p*-(trifluoromethoxy)-phenylhydrazone in K^+^-buffer Ca^2+^-free solution, and measurements were taken at both wavelengths after the fluorescence reached stable values. Then, R_max_ was obtained by saturating the indicator with 10 mM or 20 mM CaCl_2_ in the presence of 10 mM EGTA.

To calculate the K_D_, a dose response (0, 1.0, 2.0, 3.0, 4.0, 5.0, 6.0, 7.0, 8.0, 9.0, and 10.0 mM CaCl_2_/EGTA (free Ca^2+^ ranging from 0 to 37 μM)) was performed in K^+^ buffer. As a double-log plot, the Ca^2+^ response of the indicator was linear with the *x*-intercept equal to the log of the apparent K_D_ of the indicator. We calculated R_min_ = 0.66, R_max_ = 1.84, and β = 1.35 with K_D_ = 377 nM. Of note, aldosterone treatment did not change any parameters.

### 2.4. Measurement of Cation Influx Using Mn^2+^-Quenching of Fura-2 Fluorescence Quenching

The Mn^2+^ influx was measured on Fura-2/AM loaded ARVMs as described previously [11]. Fura-2 was excited at the isosbestic wavelength, 360 nm, and emission fluorescence was monitored at 510 nm. After SR depletion (see below), a 500 µM MnCl_2_ solution was perfused in the presence of NIF and KB-R7943, and the initial linear slope of the Fura-2 fluorescence’s decrease was measured, reflecting the SOC channel activity. The quenching rate of fluorescence intensity (F_360_) was estimated using linear regression of the initial decaying phase (slope, ΔF/dt) just after Mn^2+^ addition, expressed as a percentage decrease in F_360_ per min (the maximal F_360_ signal obtained before Mn^2+^ was set at 100% to correct for differences in the cell size and/or fluorophore loading). Photobleaching was <0.5%/min during measurements. All experiments were done at controlled 37 °C.

### 2.5. Electrophysiological Recordings

The *I*_SOC_ current was recorded as described previously [11]. The SR Ca^2+^ stores were depleted by successive perfusion of 5 μM Tg and 10 mM caf, and then the currents (*I*_SOC_) were elicited by a 1.2-s voltage ramp from −100 mV to +60 mV, preceded by 0 to +40 mV prepulse (400 ms) to inactivate the voltage-dependent Na^+^ channels. The *I_SOC_* were recorded before and after 5 µM BTP2 cell perfusion, and then normalized to membrane capacitance to account for cell size variations.

### 2.6. Western Blot

Western blot was performed as described previously [11] using the antibodies listed in Table 1. Briefly, ARVMs were lysed in a RIPA (radioimmunoprecipitation assay) buffer. Samples were denatured in Laemmli’s buffer; then, 20–40 μg of total proteins were loaded per lane, separated on SDS-polyacrylamide gel electrophoresis, and transferred to a polyvinylidene difluoride (PVDF) membrane. Membranes were probed overnight at 4 °C with the corresponding primary antibody. After washes, the membranes were incubated with the secondary anti-rabbit, anti-mouse, or anti-rabbit IgG 1 h at room temperature. Immunoreactive bands were detected with enhanced chemiluminescent procedure using an ECL Western Blotting Analysis System. The Western blot quantification was performed with ImageJ software with normalization of target proteins to β-actin.

### 2.7. Immunostaining

ARVMs were fixed and permeabilized with 2% paraformaldehyde (PFA)/0.5% Triton X-100 as previously described [14]. Images were acquired using a Leica TCS SP5 confocal microscope, keeping all the acquisition parameters the same at all times. For each cell, we selected a single plane at the depth where the antibody detection was optimal. After background subtraction, as semi-quantitative analysis, the total cell fluorescence signal (integrated density) of the region of interest (ROI, by circling the cells) was normalized to the ROI area of each cell using ImageJ software (1.52d, National Institutes of Health, Bethesda, MD, USA).

### 2.8. Quantitative Real-Time PCR

The RNA extraction and RT-qPCR quantification were performed as previously described [11], using the primers listed in Table 2. Briefly, total RNA from ARVMs was isolated using the TRIzol procedure according to the manufacturer’s instructions. For complementary DNA (cDNA) synthesis, 1 µg of total RNA was reverse-transcribed using the iScript cDNA synthesis kit (Bio-Rad, Hercules, CA, USA) according to the manufacturer’s instructions. Then, cDNA was used as a template for real-time PCR with SYBR Green Supermix. Quantitative determination of the different messenger RNA (mRNA) expression levels was performed with a CFX96 Touch™ Real-Time PCR Detection System with either gene-specific primers or primers for endogenous controls. The mRNA levels were normalized to housekeeping genes and were expressed as a fold change of that determined in control (ctrl) ARVMs for each isolation.

### 2.9. Statistical Analysis

The number of cells (*n*) or animals (*N*) studied per experiment is indicated. Statistical analyses, described in the figure legends, were performed using GraphPad Prism 6.0 software (GraphPad Software, San Diego, CA, USA). The variance assumption was automatically tested. A value of *p* < 0.05 was considered significant.

## 3. Results

### 3.1. Aldosterone Increases Store-Operated Ca^2+^ Entry (SOCE) via MR Activation in Adult Cardiomyocytes

We firstly examined the SOCE activity in quiescent rod-shaped Fluo-4/AM-loaded ARVMs, which were treated with or without aldosterone (Aldo) for 24 h. Of interest, 24-h aldosterone treatment did not induce cellular hypertrophy as assessed by the similar membrane capacitance of isolated ARVMs treated 24 h with or without 1 μM aldosterone (123 ± 6 vs. 120 ± 6 pF, for 23 control cells vs. 23 Aldo treated cells, respectively). As exemplified in Figure 1A, after SR depletion in a Ca^2+^-free solution containing thapsigargin (a SERCA inhibitor, Tg 5 µM), caffeine (an RyR2 activator, caf 10 mM), and nifedipine (an L-type Ca^2+^ channel blocker, NIF 10 µM), the subsequent addition of 1.8 mM Ca^2+^ in the presence of NIF and KB-R7943 (an Na^+^–Ca^2+^ exchanger inhibitor, KB 5 µM) induced a moderate increase of fluorescence in 69% of ctrl ARVMs (gray trace), while a stronger response was observed in a dose-dependent manner in all aldosterone-treated cells (100 nM, dark-gray trace and 1 µM, red trace). On average (Figure 1B), we denoted a 1.4-fold increase in Ca^2+^ entry in 1 µM aldosterone-treated ARVMs, which was prevented in co-treated ARVMs with a selective mineralocorticoid receptor (MR) antagonist RU-28318 (RU at 10 µM). Similar results were obtained using the ratiometric Fura-2/AM fluorescent probe (Figure 1C,D). To confirm those results, we used Mn^2+^-quenching microfluorimetry in the presence of major cardiac Ca^2+^ entry pathway inhibitors (NIF and KB). Figure 1E shows representative Fura-2/AM fluorescence traces after SR depletion of ARVMs treated with or without aldosterone for 24 h. The quenching rates, proportional to Mn^2+^ entry, were increased up to 1.5-fold after aldosterone incubation; the effect was blunted in ARVMs co-treated with 10 μM RU-28318 (Figure 1F). These data confirmed an increased cation entry after aldosterone treatment, which was dependent on MR activation.

To further explore the nature of the SOCE modulated by the aldosterone pathway, we recorded the store-dependent currents using the patch-clamp technique in whole-cell configuration, using solutions to limit Ca^2+^ and K^+^ currents by including inhibitors (NIF and KB) and Cs^+^ to block K^+^ channels. After SR Ca^2+^ store depletion (see above), the SOC currents (*I*_SOC_) were elicited by a standard ramp protocol before and after the perfusion of the non-selective SOC channel blocker BTP2 (5 µM). As shown in Figure 2A–C, the small BTP2-sensitive currents with a linear current density–voltage relationship and a reversal potential around 0 mV were recorded in ARVMs incubated without aldosterone. The *I*_SOC_ was increased in aldosterone-treated ARVMs (Figure 2C). The feature of this aldosterone-enhanced *I_SOC_* induced by store depletion, i.e., its reversal potential around 0 mV and sensitivity to BTP2, suggested the activation of non-selective cationic channels carried by TRPCs.

### 3.2. Aldosterone Increases TRPC1, TRPC5, and STIM1 Protein Expression in Adult Cardiomyocytes

To assess the molecular nature of the aldosterone-increased SOCE, we investigated the effect of aldosterone treatment on the expression of different SOCE machinery actors, i.e., TRPCs and Orai1 channels and STIM proteins in ARVMs after 24 h of incubation. As assessed by Western blots (Figure 3), we observed an increased protein expression of TRPC1, TRPC5, and STIM1, without any changes in Orai1, TRPC3, TRPC4, TRPC6, STIM2, and Orai3 protein levels. Uncropped blots are presented in Appendix A. The increases in TRPC1, TRPC5 and STIM1 protein expressions were further tested by immunostaining of treated isolated cardiomyocytes (Figure 4). The fluorescence signal of these three proteins appeared more intense in 1 µM Aldo-treated cells than in ctrl cells and revealed a localization of TRPC1 and STIM1 at the surface membrane, while TRPC5 appeared to be present at the striated area, as previously reported [2,28,29,30]. At mRNA levels, using RT-qPCR, we observed an increased mRNA expression of TRPC1 and TRPC5 induced by 100 nM and 1 µM aldosterone treatment, whereas lower doses (10 nM) had no effect. STIM1 mRNA levels remained unchanged after 24 h of aldosterone treatment, no matter the concentration used (Figure 5).

## 4. Discussion

Our study was aimed at investigating the aldosterone effect on SOCE in ARVMs. Our results indicate that a 24-h treatment of high-dose aldosterone stimulates the SOCE in an MR-dependent manner, due to a specific upregulation of TRPC1, TRPC5, and STIM1.

Upregulation of the cardiac mineralocorticoid signaling pathway emerged as a key regulator involved in heart failure development and progression [31], in which Ca^2+^-dependent activation of transduction signaling is also important [32]. On the other hand, the re-emergence of cardiac SOCE machinery was recently shown in the heart failure pathological remodeling processes, contributing to Ca^2+^ prohypertrophic signaling [1,3,4]. Notably, the TRPC1 channel, which is the major isoform in whole human heart [33], is upregulated during heart failure progression and is critical for it [17,22,34,35,36,37]. Likewise, increased TRPC5 expression in human heart failure [38,39] and STIM1 elevation are linked to cardiac hypertrophy [9,40,41,42,43]. However, the mechanism regulating cardiac TRPC channels expression is not well defined. Since cardiac hypertrophy and heart failure can be viewed as a gene regulatory disorder [44], we hypothesize that MR as transcription factor might be involved in this process.

As previously reported [14], ARVMs show rather small Ca^2+^ entry following SR depletion, which is enhanced after aldosterone treatment. The presence of SOCE in adult ventricular cardiomyocytes first reported in 2004 [13] remains controversial. For example, the traditional protocol using only thapsigargin or cyclopiazonic acid (CPA) to block the SR Ca^2+^ pumps indicates that there is no or limited SOCE in normal adult ventricular cells [9,17,35,45]. However, these conditions might not sufficiently deplete the SR Ca^2+^ stores of quiescent adult ventricular cells [15,46], even if one might think that, in the presence of an SERCA pump inhibitor, the SR Ca^2+^ leak would deplete the SR [45]. Thus, when a clear SR Ca^2+^ depletion is observed (with ryanodine receptor activation through either stimulation or caffeine/ryanodine application), a moderated SOCE activity might be observed [11,14,47,48,49], as we show in the present study.

Reflecting previous reports in adrenal chromaffin cells [50] or coronary arteries [51] from metabolic syndrome pigs, in the mesenteric arterial smooth muscle cells of DOCA-salt hypertensive rats or in A7r5 cells [52] and in NRVMs [11], a chronic treatment of ARVMs with aldosterone potentiates SOCE, in line with the emergence of a marked BTP2-sensitive current after store depletion. This linear-shaped non-selective current, as assessed by its reversal potential near 0 mV, features reminiscent of TRPC channels [36,53,54]. Indeed, SOCE is a ubiquitous mechanism that is mediated by distinct SOC channels, ranging from the highly selective Ca^2+^ release-activated Ca^2+^ channel supported by the Orai1 channel to relatively Ca^2+^ non-selective SOC channels supported by TRPC channels [55]. Unlike NRVMs, in which aldosterone treatment enhances Orai1 expression [11], we did not observe an upregulation of Orai1 in ARVMs. This points out that the use of immature neonatal cardiomyocytes presenting unequivocal robust SOCE [9,10] might have limitations, thus justifying the present study. Nonetheless, aldosterone promotes an MR-specific increase in TRPC1/5 and STIM1 protein expression in ARVMs, as seen in metabolic syndrome adrenal chromaffin cells [50], in coronary arteries [51,56], and in NRVMs [11]. These are consistent with the ionic current features that we observed. Aldosterone increases STIM1 expression only at the protein level, suggesting a post-transcriptional regulation. In this way, it was demonstrated that STIM1 protein expression is regulated by the serum- and glucocorticoid-inducible kinase 1 (SGK1), an aldosterone-regulated kinase which phosphorylates Nedd4-2, leading to decreased ubiquitination and subsequent degradation of the STIM1 protein [57,58]. A number of studies, conducted in various cell models, implicated members of TRPC channels in SOC activity [59,60,61,62]. Notably, TRPC1 channels are believed to mediate the non-selective cation current and to form SOC channels as a component in human atria [63], in the mouse sinoatrial node [64], in human cardiac c-kit+ progenitor cells [65], in NRVMs [66], and in adult ventricular cardiomyocytes [36,46,67]. Similarly, TRPC5, which was found to contribute to the formation of SOCE in smooth muscle cells isolated from rabbit pial arterioles [68], contributes to SOCE in NRVMs [11,69]. In addition, STIM1, the critical constituent of SOCE, can directly bind to and activate TRPC1 and TRPC5 channels via interactions in its ezrin/radixin/moesin (ERM) domain [70,71,72,73,74,75,76].

Our findings provide direct support for the presence of STIM1, TRPC1, and TRPC5 in ARVMs and for their participation in aldosterone/MR-enhanced SOCE. Taking into account that G-protein-coupled receptor (GPCR) activation promotes TRPC1 and TRPC5 translocation to the plasma membrane in a STIM1-dependent manner [77] and the documented signaling crosstalk between cardiac MR and GPCR signaling components [78], further studies directed toward the potential of the cardiac MR-dependent TRPC channelosome upregulation in the receptor agonist (such as angiotensin II)-induced inositol 1,4,5-trisphosphate (IP_3_) sensitive Ca^2+^ store depletion remain to be performed.

## Figures and Tables

**Figure 1 cells-09-00047-f001:**
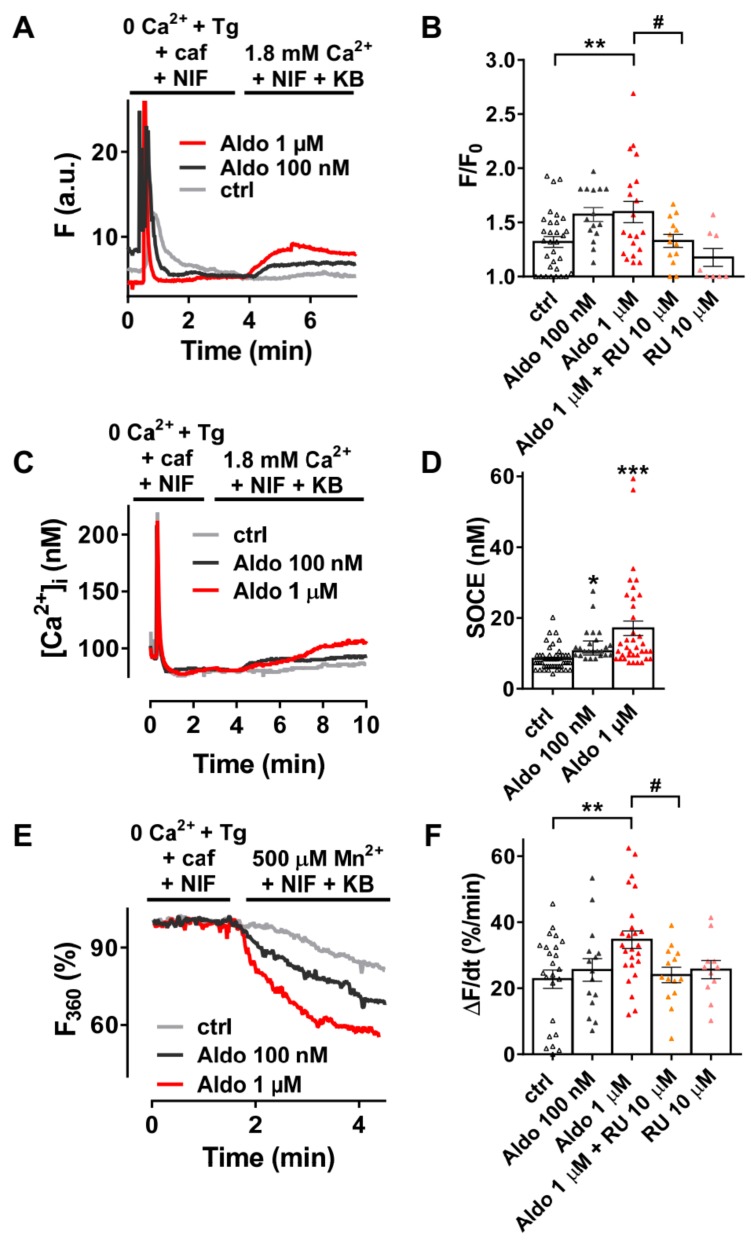
A 24-h aldosterone treatment increases store-operated Ca^2+^ entry (SOCE) in adult rat ventricular cardiomyocytes (ARVMs). (**A**) Representative traces of fluorescence variation in Fluo-4/AM-loaded ARVMs pre-incubated in the absence (control (ctrl), gray trace) and in the presence of aldosterone (Aldo, 100 nM, dark-gray trace or 1 µM, red trace). Cells were exposed to 5 µM thapsigargin (Tg) plus 10 mM caffeine (caf) in the presence of 10 µM nifedipine (NIF) in Ca^2+^-free medium to deplete the SR and then to Ca^2+^-containing solution in the presence of 5 µM of KB-R7943 and NIF to evaluate the SOCE. (**B**) Averaged amplitude of SOCE (F/F_0_) in ARVMs incubated for 24 h in the absence (ctrl, white symbols) or in the presence of aldosterone (100 nM, gray symbols or 1 µM, red symbols) without or with 10 µM of the selective MR antagonist RU-28318 (RU, orange symbols in aldosterone condition and pink symbols in ctrl condition). Average results of 8–32 cells from three isolations. (**C**) Representative traces of [Ca^2+^]_i_ variation (in nM) in Fura-2/AM-loaded ARVMs incubated for 24 h in the absence (ctrl, gray trace) or in the presence of aldosterone (100 nM, dark-gray trace; 1 µM, red trace). Fura-2/AM fluorescence 340/380 ratios were converted to [Ca^2+^]_i_ as described in the methods. (**D**) Quantitative assessment of SOCE (in nM) from untreated (ctrl, white symbols) and 100 nM (gray symbols) or 1 µM (red symbols) aldosterone-treated cells. Average results of 24–48 cells from four isolations. (**E**) Representative traces of Mn^2+^-induced Fura-2/AM fluorescence decay in ARVMs incubated in the absence of aldosterone (ctrl) are shown in gray traces, and those in the presence of aldosterone 100 nM or 1 µM are shown in dark-gray and red traces, respectively. (**F**) Bar graph of the initial slope of the Mn^2+^-induced decrease of Fura-2/AM fluorescence fitted by linear regression and averaged (Δ*F*/d*t*, %/min) for ARVMs incubated for 24 h in the absence (ctrl, white symbols) or in the presence of aldosterone (100 nM, dark-gray symbols or 1 µM, red symbols) without or with 10 µM RU (orange symbols in aldosterone condition and pink symbols in ctrl condition). Average results of 11–25 cells from three isolations. Results are presented with scatter plots with mean ± standard error of the mean (SEM). Statistical significance was evaluated using one-way ANOVA followed by post hoc Fisher least significant difference (LSD) test for multiple comparisons. * *p* < 0.05, ** *p* < 0.01, *** *p* < 0.001 vs. ctrl; ^#^
*p* < 0.05 vs. aldosterone.

**Figure 2 cells-09-00047-f002:**
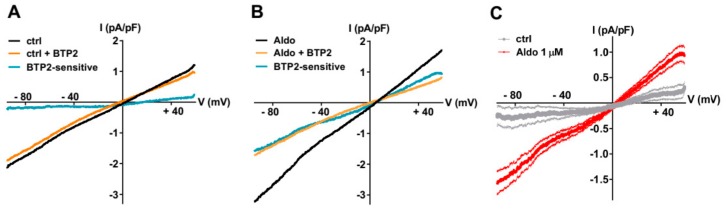
A 24-h aldosterone treatment increases SOC currents (*I*_SOC_) in ARVMs. (**A**,**B**) Representative current–voltage (I–V) relationships elicited by ramp voltage-clamp protocol in ARVMs incubated for 24 h without (**A**) or with aldosterone 1 μM (**B**), following SR Ca^2+^ depletion (black traces) and then in the presence of 5 µM BTP2 (orange traces). BTP2-sensitive (blue traces) represents the difference current. (**C**) Scatter plots of mean ± SEM of the BTP2-sensitive I–V relationships recorded in ARVMs incubated for 24 h without (gray trace, −0.19 ± 0.19 pA/pF at −90 mV and 0.30 ± 0.17 pA/pF at +60 mV, *n* = 7) or with Aldo 1 µM (red trace, −1.51 ± 0.20 pA/pF at −90 mV (*p* < 0.01) and 0.90 ± 0.15 pA/pF at +60 mV (*p* < 0.05), *n* = 9) from three cell isolations. Statistical significance was evaluated using Student’s *t*-test.

**Figure 3 cells-09-00047-f003:**
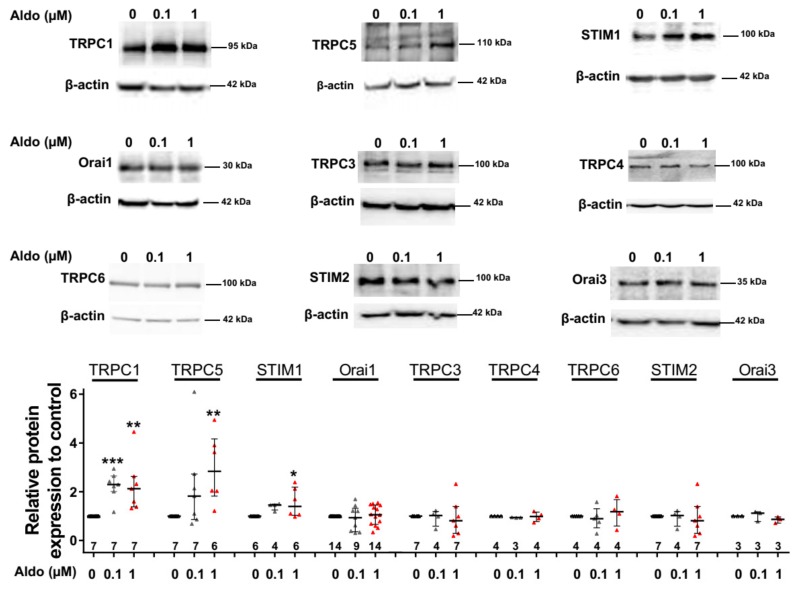
A 24-h aldosterone treatment increases transient receptor potential canonical channel 1 (TRPC1), TRPC5, and STIM1 protein expression in ARVMs. Representative Western blot (top panels) and pooled data (bottom bar graph) of TRPC1, TRPC5, STIM1, Orai1, TRPC3, TRPC4, TRPC6, STIM2, and Orai3 in ARVMs incubated for 24 h in the absence or in the presence of aldosterone (100 nM or 1 µM). Protein levels were normalized by β-actin and are expressed as fold changes of that determined in untreated ARVMs for each cell isolation. Results are presented as scatter plots with median and interquartile range. Statistical significance was evaluated using one-way ANOVA (Kruskal–Wallis) followed by post hoc Dunn’s test for multiple comparisons. ** p < 0.05*, ** *p* < 0.01, *** *p* < 0.001 vs. ctrl.

**Figure 4 cells-09-00047-f004:**
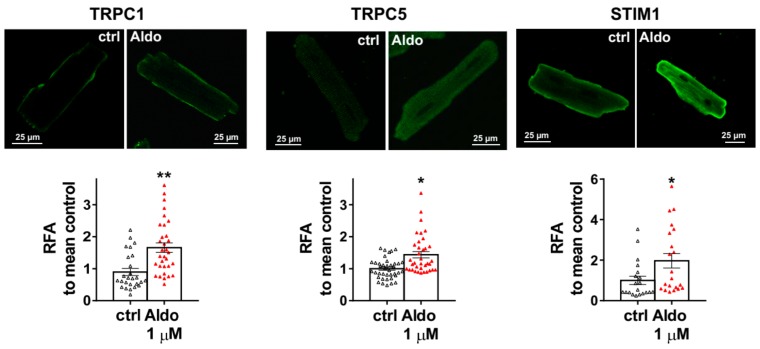
Cellular localization of TRPC1, TRPC5, and STIM1 in ARVMs. Representative immunofluorescences images (top) and quantification of TRPC1, TRPC5, and STIM1 in ARVMs incubated for 24 h in the absence or in the presence of aldosterone 1 µM. Results are expressed as relative fluorescence/cellular area (RFA). Average results of 20–39 cells from three isolations. Results are presented as scatter plots with mean ± SEM. Statistical significance was evaluated using Student’s *t*-test. * *p* < 0.05, ** *p* < 0.01 vs. ctrl.

**Figure 5 cells-09-00047-f005:**
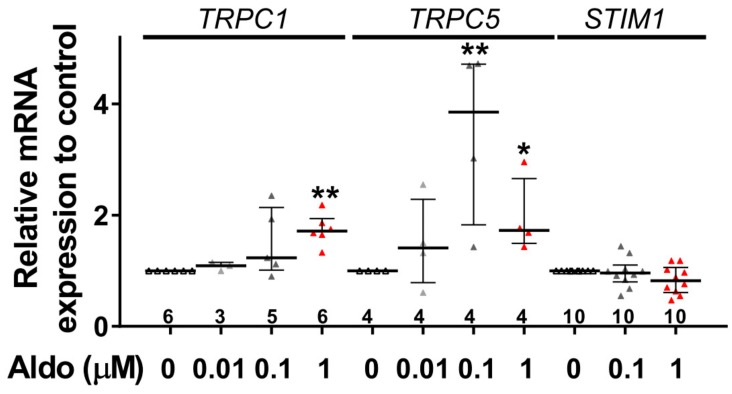
A 24-h aldosterone treatment increases *TRPC1* and *TRPC5* messenger RNA (mRNA) expression in ARVMs. Relative *TRPC1*, *TRPC5*, and *STIM1* mRNA levels in ARVMs were incubated for 24 h in the absence or in the presence of aldosterone (Aldo 10 nM, 100 nM or 1 µM). The mRNA levels determined by qRT-PCR were normalized to housekeeping genes and are expressed as fold changes of that determined in ARVMs incubated in the absence of aldosterone for each cell isolation. Results are presented as scatter plots with median and interquartile range. Statistical significance was evaluated using one-way ANOVA (Kruskal–Wallis) followed by post hoc Dunn’s test for multiple comparisons. * *p* < 0.05, ** *p* < 0.01 vs. ctrl.

**Table 1 cells-09-00047-t001:** List of primary antibodies. TRPC—transient receptor potential canonical channel.

Protein	Host	Dilution	Code	Source
TRPC1	Mouse	1/200	SC-133076	Santa Cruz
TRPC3	Rabbit	1/200	ACC-016	Alomone
TRPC4	Rabbit	1/200	ACC-018	Alomone
TRPC5	Mouse	1/200	73-104	UC Davis/NIH NeuroMab Facility
TRPC6	Rabbit	1/200	ACC-017	Alomone
Orai1	Rabbit	1/200	O8264	Sigma
Orai3	Rabbit	1/200	ACC-065	Alomone
STIM1	Rabbit	1/200	S6197	Sigma
STIM2	Rabbit	1/200	ACC-064	Alomone
β-actin HRP	Mouse	1/30,000	SC-47778	Santa Cruz

**Table 2 cells-09-00047-t002:** List of primers used for RT-qPCR.

Gene	Forward (5′–3′)	Reverse (5′–3′)
*RPL32*	GCT GCT GAT GTG CAA CAA A	GGG ATT GGT GAC TCT GAT GG
*YWHAZ*	AGA CGG AAG GTG CTG AGA AA	GAA GCA TTG GGG ATC AAG AA
*TBP*	AAA GAC CAT TGC ACT TCG TG	GCT CCT GTG CAC ACC ATT TT
*STIM1*	TCT CTG AGT TGG AGG ATG AGT AGA	CAA TAT AGG GGA GCA GAG GTA AGA
*TRPC1*	TTC CAA AGA GCA GAA GGA CTG	AGG TGC CAA TGA ACG AGT G
*TRPC5*	TGA GTG GAA GTT TGC GAG AA	TGG GAC AGA AGG TGT TGT TG

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
