# Peer review of "Specific Upregulation of TRPC1 and TRPC5 Channels by Mineralocorticoid Pathway in Adult Rat Ventricular Cardiomyocytes"

_cells, 2019, doi:10.3390/cells9010047_

Round 1

Reviewer 1 Report

The manuscript of Bartolo F. et al. describe the increase expression to TRPC1 and TRPC5 channels upon 24h aldosterone stimulation in adult rat ventricular cardiomyocytes. The manuscript is well written and the results support well the conclusions. 

I have no specific comments on the manuscript and support its publication in Cells. 

Author Response

We would like to thank the reviewer for its positive evaluation of our manuscript.

Reviewer 2 Report

In this short communication paper, the authors found the that the TRPC1 and 5 are upregulated in the adult rat ventricular cardiomyocytes by the activation of mineralocorticoid pathway. This is a very interesting tiopic, however, there are couple questions that the authors still need to address.

Comments:

There is no introduction about mineralocorticoid pathway, either about the aldosterone, this should be included. This would help to validate the methods, like incubation for 24 hours. 24 hours culture may introduce various changes and remodeling to the acutely isolated cardiomyocytes. Is there any specific culture method applied here to avoid such changes? This should be detailed in the methods. Does the culture for 24 hours affect the TRPC expression and function? The authors should add the control group on freshly isolated cells to compare with the 24 hours incubation cells. In Figure1, the authors use Fluo-4 fluorescent intensity comparison to draw the conclusion, however, this method is not quantitative, ratiomatric dye should be considered with careful calibration.  The authors need to show a statistic IV curve in Figure 1E. Fresh isolated cells should be added as control in the PCR experiments. Detailed methods on the immunofluorescent normalization should be provided, as this method is also not quantitative. So the conclusion should be carefully drew.

Author Response

see attach file

Reviewer 3 Report

The present study investigates the role of aldosterone in the modulation of SOCE by upregulating the expression of TRPC1 and TRPC5. The authors show that aldosterone enhances TG+caffeine-induced SOCE and Mn2+ entry and the electrophysiological recordings are compatible with an increase in the ISOC current. All these findings point to the involvement of TRPC channels in the effect of aldosterone. Further studies reveal that TRPC1 and TRPC5 are upregulated upon treatment for 24 h with aldosterone. This is a short communication about an isssue that is novel and interesting. The study is nicely designed and the manuscript is well written.

There are a few points that need to be addressed:

It would be relevant to analyse the effect of aldosterone itself on Ca2+ mobilisation. Is the upregulation of TRPC1 and TRPC5 induced after treatment for 24 h with aldosterone reversible?

Author Response

Response to the reviewer #3

The present study investigates the role of aldosterone in the modulation of SOCE by upregulating the expression of TRPC1 and TRPC5. The authors show that aldosterone enhances TG+caffeine-induced SOCE and Mn2+ entry and the electrophysiological recordings are compatible with an increase in the ISOC current. All these findings point to the involvement of TRPC channels in the effect of aldosterone. Further studies reveal that TRPC1 and TRPC5 are upregulated upon treatment for 24 h with aldosterone. This is a short communication about an isssue that is novel and interesting. The study is nicely designed and the manuscript is well written.

There are a few points that need to be addressed:

It would be relevant to analyse the effect of aldosterone itself on Ca2+ mobilisation.

Whether a rapid non-genomic effect of aldosterone has been described in various cell types, we never observed such effect on cardiomyocytes (Benitah et al. Circ Res, 1999). Indeed, in our previous study (Sabourin et al. JBC, 2016, Supplemental figure 2), we reported that there was no effect of direct aldosterone perfusion on SOCE from NRVMs.

Is the upregulation of TRPC1 and TRPC5 induced after treatment for 24 h with aldosterone reversible?

We already provide that the aldosterone effect can be blunted in presence of mineralocorticoid receptor antagonist (new Figure 1). The reversibility of chronic aldosterone treatment is challenging since discontinued treatment suppose the ARVMs maintenance for at least 48 hours in vitro with functional and structural integrities, which we did not reach yet.

Round 2

Reviewer 2 Report

The reviewer would thank the authors for the reply, and has no further questions.